# Large-Scale Mercury Dispersion at Sea: Modelling a Multi-Hazard Case Study from Augusta Bay (Central Mediterranean Sea)

**DOI:** 10.3390/ijerph19073956

**Published:** 2022-03-26

**Authors:** Mario Sprovieri, Andrea Cucco, Francesca Budillon, Daniela Salvagio Manta, Fabio Trincardi, Salvatore Passaro

**Affiliations:** 1Consiglio Nazionale delle Ricerche, Istituto per lo Studio Degli Impatti Antropici e Sostenibilità in Ambiente Marino, 91021 Trapani, Italy; mario.sprovieri@cnr.it (M.S.); daniela.salvagio@ias.cnr.it (D.S.M.); 2Consiglio Nazionale delle Ricerche, Istituto per lo Studio Degli Impatti Antropici e Sostenibilità in Ambiente Marino, 09170 Oristano, Italy; andrea.cucco@cnr.it; 3Consiglio Nazionale delle Ricerche, Istituto di Scienze Marine, 80133 Napoli, Italy; francesca.budillon@cnr.it; 4Dipartimento di Scienze del Sistema Terra e Tecnologie per l’Ambiente, 00185 Rome, Italy; fabio.trincardi@cnr.it

**Keywords:** hazard chain, mercury contamination, earthquake damages, numerical modelling, marine-coastal area

## Abstract

This contribution discusses an example of potential multi-hazard effects resulting from an earthquake in a highly seismogenic area of the Mediterranean Sea, the Augusta Bay, which presents high levels of contamination in sediments and seawater, due particularly to high-concentrations of mercury as a result of a long-term industrial exploitation. In particular, a high-resolution hydrodynamic and transport model is used to calculate the effects of enhanced mercury spreading in the open sea after significant damage and collapse of the artificial damming system confining the embayment where a very high concentration of Hg occurs in seafloor sediments and seawater. Coupling high-resolution 3D dynamic circulation modelling and sediment–seawater Hg fluxes calculated using the HR3DHG diffusion–reaction model for both inorganic and organic Hg species offers a valuable approach to simulating and estimating the effects of spatial dispersion of this contaminant due to unpredictable hazard events in coastal systems, with the potential attendant enhanced effects on the marine ecosystem. The simulated scenario definitely suggests that a combination of natural and anthropogenic multi-hazards calls for a thorough re-thinking of risk management in marine areas characterised by significant levels of contamination and where a deep understanding of the biogeochemical dynamics of pollutants does not cover all the aspects of danger for the environment.

## 1. Introduction

The fast development of industrialised countries has rapidly affected marine and coastal areas through installation of industrial plants and, in particular, petroleum refineries and related chemical plants. This trend has led to significant environmental degradation that requires drastic solutions and innovative approaches to ecosystems and environmental recovery. Indeed, highly contaminated coastal and marine areas and their specific compartments such as sediments, seawater, groundwater, etc. represent a direct and primary hazard for the environment, the ecosystem, and the well-being of human populations (e.g., [1,2]).

The dynamics and specific kinetics of biogeochemical cycles of organic and inorganic contaminants in all environmental matrices are key priority issues of investigation in any assessment of ecosystem status. Nonetheless, this traditional approach does not consider the geological risk components associated with earthquakes, landslides, tsunamis, etc., and their potential triggering effect, mainly in contaminated areas, potentially leading to amplified chains of hazard. A dramatic example comes from the Mw-9.0 earthquake of 11 March 2011 at Tohoku (Japan), one of the largest on record, which caused major damage in terms of loss of human life and destruction of buildings and infrastructure. Following the main shock, a 11.5–15.5-metre-high tsunami wave struck the Daiichi area [3]. This event is principally remembered worldwide for the serious damage to the Fukushima Daiichi Nuclear Power Station, which caused a core meltdown and hydrogen explosions. The Fukushima event and its consequences evidence two main needs in terms of understanding of a multi-hazard approach: (i) a detailed knowledge of the short- to long-term geological history of an area in order to more realistically define the recurrence potential of major convulsive events; and (ii) the importance of taking into account the multiple natural risks associated with earthquakes, tsunamis, flash flooding, coastal slides, etc. [4,5,6]. These natural phenomena can generate distinctive damage that in turn may activate much more complex chains of hazards. This demands new vision on the issue of multi-hazards based on consistent hypotheses concerning nested sequential events, both natural and anthropogenic, that may lead to a non-linear sequence of successive negative events (e.g., [7]). In this context, Agenda 21 (UNCED 1992) underlined the need to fill gaps in knowledge to properly manage combined natural and anthropogenic events in areas where the risk of natural disasters (UNCED 1992, paragraph 7.61) appear more probable.

The Mediterranean Basin is threatened by high geological risk because of its young age and complex geologic and geodynamic setting. In addition, a relevant number of highly contaminated areas punctuate the marine and coastal areas of the Mediterranean. The combination of these natural and anthropogenic components makes the Mediterranean Sea highly vulnerable to multi-hazards. However, a rather limited understanding and investigation is currently available concerning this combination of potential effects.

This work presents an example of how geological hazards, specifically deriving from an earthquake, may lead to large-scale dispersion of mercury from human activities in a semi-enclosed costal bay, Augusta Bay in Sicily, Southern Italy. Owing to multi-contaminant discharge which occurred during the 20th century, Augusta Bay has a very high pollution level and is included in the National Remediation Plan by the Italian Environmental Ministry (Law 426/98, Ministerial Decree 10 January 2000). Earthquakes, tsunamis, exceptionally strong sea-storms (e.g., Rapallo, Northern Italy on 30 October 2018 [8]) and/or operational problems linked to substandard ships and poor shipping practices (e.g., [9]) may result in partial damage to or disruption of the bay seawalls. The Augusta Harbor breakwater system is made up of three segments composed of chaotically arranged concrete blocks, with the exception of the northernmost one, which is characterized by three sections (from north to south, respectively) composed of (i), A concrete seawall about 400 m length; (ii) a sequence of neatly arranged blocks ca. 400 m length, and (iii) chaotically arranged concrete blocks spread over more than 1100 m (see Appendix A).

In this work, we simulate the effects of a potential complete collapse of the Augusta seawalls (the artificial breakwater system enclosing the bay) under external stresses.

Such an extreme event could occur along the eastern Sicilian coast; this region has experienced a number of seismic events considered among the strongest ever observed in Italy in the historical past (e.g., [10]), and these have occasionally generated tsunamis (see [11] and references therein). A comparable effect with a damming collapse occurred in the port of Rapallo in 2018, triggered by an enormous swell (see: https://www.youtube.com/watch?v=andOXmU5tvc, accessed on 7 January 2020).

What happens if a natural disaster hits the Augusta Bay area, as has happened in historical past in this area [10,11,12] with tsunamis and seismic events? In this paper, we analyse the potential environmental impacts that may be induced by the fall of a confining seawall and the consequent outflow of mercury from the harbor and into the adjacent continental shelf and open marine environment. We use a high-resolution hydrodynamic and transport model capable of taking into account the contaminants that have already been measured and modelled in terms of their biogeochemical behaviour in the study area in order to evaluate the potential risk of contamination due to the massive spread of Hg in the event of a breached or collapsed seawall at Augusta Bay.

## 2. The Case Study: Augusta Bay

Augusta Bay is a natural semi-enclosed marine area located in Eastern Sicily on the Ionian Sea in Southern Italy (Figure 1a). Two peninsulas, one stretching southward and the other linked to the land by a sand spit, separate the bay into three inlets, Xifonio, Megarese, and Priolo. The largest portion of the bay, the “Rada di Augusta” (~23.5 km^2^) is delimited in the northern sector by the City of Augusta and to south and east by an artificial seawall built in 1970 and made of cubic blocks of ~2 m^3^ grounded in 2 m of water (Figure 1b). This area comprises the Augusta harbor, one of the most important commercial ports in Europe, which is a military and industrial facility with intensive naval traffic. From the 1960s until the present the bay has hosted several large industrial plants, including oil refineries and chemical and petrochemical industries, which have substantially harmed the surrounding marine environment. In particular, an important chlor-alkali plant operated from 1958 to 2005, using mercury cells and discharging wastewater without pre-treatment into the bay until the late 1970s [13,14,15]).

The water circulation in the area is characterized by the Messina Rise Vortex [16] and by a marked seasonal dynamic, with a main southward coastal current transporting the surface waters from the Ionian Sea towards the Sicily Channel. Given the small extent of the continental shelf in the study area (Figure 1b; see Firetto Carlino et al. [17]), this current directly influences the transport dynamics in coastal waters [18]. The Mistral from the NW, Greek from the NE, and Sirocco from the SW are the dominant winds in the area, with intensities up to 10 m/s during the late autumn and winter [19]. The tidal component is semidiurnal, lunar component M2 alone, with an amplitude of less than 20 cm [20].

**Figure 1 ijerph-19-03956-f001:**
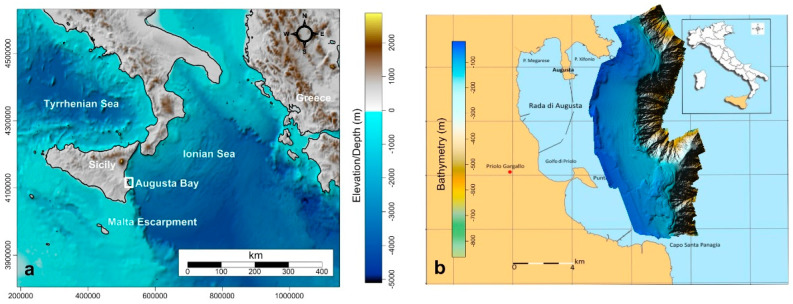
(**a**) Location map of Augusta Bay; (**b**) coastal toponyms of Augusta Bay and of the Rada di Augusta. Bathymetry in B modified after Firetto Carlino et al. [17].

## 3. Environmental Setting

### 3.1. Geological and Morphological Framework

The study area is located on the northernmost sector of the Malta Escarpment [21], a tectonically active morpho-structural lineament (e.g., [22,23]; Figure 2a). The E sector of Sicily includes the Peloritani Mountains and Mount Etna, the highest European volcano at 3328 m above the sea level. The NE–SW-striking Catania–Gela foredeep and the Hyblean plateau and foreland represent the relatively less deformed sector of the collision zone [24], which is characterized by various tectonic regimes and earthquake focal mechanisms. In detail, NW–SE strike–slips are prevalent in the Sicily Strait, while prevailing NNW–SSE lineaments rules run from the Aeolian Islands to Mount Etna, continuing southward along the northernmost prolongation of the Malta Escarpment [25,26,27]. The Malta Escarpment is a major morpho-structural lineament consisting of a regional, NNW–SSE-trending fault system offshore of Sicily [26]. Several tectonic lineaments along the eastern flank of Sicily are considered seismogenic sources. In the Augusta Bay, in particular, earthquakes have been recorded in recent times with an average weighed magnitude of seven or greater based on their macro-seismic intensity field [10,27,28,29,30].

The Augusta onshore shows outcrops of Oligo-Miocene wackestone and packstone unconformably overlying the Cretaceous-Eocene shelf margin and slope carbonate and submarine volcanoclastic beds. Several NW–SE horsts in the coastal sector (the Monte Tauro, Augusta, Magnisi, S. Panagia, and Maddalena horsts) are bounded by structural depressions filled with Pleistocene bioclastic and detrital deposits, forming the modern alluvial plains. Fluvial and alluvial, swamps, ponds, and salt flats dominate the coastal depositional environments from the Late Pleistocene up to recent times until industrial settlement. In the area surrounding Augusta, a coastal terrace ascribed to the Eemian sea-level high stand is markedly displaced vertically, between +10 m and −10 m relative to modern sea level, indicating significant tectonic activity in the area over the last 100 ky [17].

Eastern Sicily is the European sector with the largest record of tsunami impacts in historical times (e.g., [30] and references therein). Four main events, dated to 1169 AD, 1542 AD, 1693 AD, 1908 AD, have been largely reconstructed starting from the assumption of “near-field” sources, i.e., offshore faults or other fault segments located near the Sicilian coasts, although their precise location is debated [31]. Several other tsunami events have been reconstructed based on historical artifacts and depositional sequences [32]. The Augusta coastal area lies close by to a seismically active zone which has generated several earthquakes over the last five centuries, causing numerous casualties and severe destruction of villages. Historical documents report detailed descriptions of three tsunamis triggered by the major earthquakes of 1542, 1693, and 1908 A.D., which enhanced the destruction induced by the earthquakes [33]. In addition, official data published by Italian earthquake monitoring centers [28] have reported more than seventy major events (equivalent Maw > 5.0) in the last 400 years, and more than ten earthquakes with equivalent Maw > 6.0. The last major event recorded in historical times occurred in 1693, with an equivalent Maw > 7.5 [12] (Figure 2b).

The high seismicity and recurrence of dramatic historical events documented in the area indicate that a strong earthquake hitting the area could destroy the system of seawalls protecting the Rada di Augusta, resulting in the potential dispersion of contaminants across the shelf and into the open sea and cascading effects on living resources.

### 3.2. Environmental Pollution

Augusta Bay represents one of the Mediterranean coastal and marine areas most pervasively affected by historical industrial pollution. Specifically, concentrations of inorganic and organic contaminants higher than threshold values for the Italian and European Legislation (e.g., Decree Law 172/2015, Directive 2008/56EC) were recorded in the bottom sediment [13,34,35,36,37] as well as in the most superficial levels (0–10 cm) (Figure 3).

Historical pollution in Augusta Bay poses an effective threat to the marine ecosystem and the fishery market in this zone with potential consequences for human health [38,39,40,41]. Salvagio Manta et al. [42] recorded active Hg transfer from sediments to seawater using an in situ benthic chamber and estimated a mercury benthic flux of about 1.3 ± 0.1 kmol y^−1^ for the entire bay, emphasizing the role of the bottom sediment as a “secondary” source of this pollutant for the overlying water column. High Hg concentrations, up to three orders of magnitude higher than the background value estimated for the Mediterranean seawater (0.1–0–8 ng L^−1^; [43,44,45,46]), have been measured in seawater in Augusta Bay [40]. The biogeochemical dynamics of the three Hg species, Hg^0^, Hg^II^, and Methyl-Hg (MeHg) as well as total Hg (Hg_T_) in Augusta Bay were recently explored by a high-resolution 3D model (HR3DHG) which combined (i) diffusion–reaction equations for estimating dissolved mercury in sediment pore water and (ii) a sorption/de-sorption process for measuring total mercury in the sediments [19]. Specifically, the spatial–temporal variability of dissolved and total mercury concentration in both seawater and upper sediment layers (0–5 cm beneath sea floor) as well as the Hg fluxes at the boundaries of the 3D model domain were theoretically reproduced and compared to the experimental data reported in the literature [36,42] from the same area. Synthetically, the experimental and theoretical results indicate that the amount of total Hg- bound to the particulate matter represents ~47% of Hg_T_ in seawater. The dissolved Hg (Hg_D_) covers about 35% of the total budget of Hg_T_ in seawater, while Hg_D_ in pore water represents a minor fraction of total Hg in the sediments [19]. The estimated Hg^II^/Hg_T_, Hg^0^/Hg_T_, and MeHg/Hg^T^ ratios are ~79%, 18%, and 2.5%, respectively, which is in good agreement with values reported from other contaminated sites (e.g., Canu et al. [47]). The same ratio has been measured for mercury species which outflow from the two inlets of Augusta Bay to the open sea. The elemental mercury concentration at the surface contributes to the mercury evasion flux, although the estimated values do not significantly influence the budget of Hg in seawater [48]. On the whole, mercury dissolved in seawater derives from sediments through the benthic flux of Hg^II^ and MeHg. In particular, these two mercury species are released directly by the sediments, while Hg^0^ is generated by redox reactions which involve the other two species. The elemental mercury concentration at the water surface contributes to the mercury evasion flux even if only a small part of elemental mercury in the seawater is released in the atmosphere [48]. The modelling results of HR3DHG demonstrate the crucial role of recycling processes in the mercury mass balance of Augusta Bay, that most (94%) of the amount of mercury released by sediments remains within the Augusta basin, and that mercury outflows at the boundaries of the basin are an order of magnitude lower with respect to the annual benthic mercury fluxes. Furthermore, the dynamics of the particulate matter deposition–resuspension process [49,50] does not significantly appear to modify the spatial distribution of the HgT recycled at the surface layer of the sediments.

## 4. Methods

The potential risk due to massive spreading of Hg triggered by collapse of the damming system confining Augusta Bay was evaluated using a numerical modelling approach.

### 4.1. The Numerical Model

A high-resolution three-dimensional ocean model, SHYFEM [51], was applied to reproduce the main hydrodynamics in Augusta Bay and the surrounding coastal areas. SHYFEM is an open-source ocean modelling software (https://github.com/SHYFEM-model/shyfem, accessed on 1 May 2020) based on the finite element method and has been applied to dozens of settings worldwide, including lagoons, coastal areas, shelf areas, and open seas, with the aim of reproducing water circulation, surface wave propagation, sediment transport, pollutant transport, etc. Most of these applications have focused on the Mediterranean Sea through operational forecasting of the main ocean features at the basin [52], sub-basin [53,54,55,56], and coastal scales [18,57].

The model resolves the 3D shallow water equations vertically integrated over z-layers. The spatial discretization of the unknowns is carried out with the finite element method, whereas a semi-implicit algorithm is used for integration in time [51]. SHYFEM accounts for barotropic, baroclinic, and atmospheric pressure gradients as well as wind drag and bottom friction, non-linear advection, and vertical turbulent processes [51]. The solved equation system for a single layer l reads as follows:(1)$$\begin{array}{l} \frac{\partial U_{l}}{\partial t}+u_{l}\frac{\partial U_{l}}{\partial x}+v_{l}\frac{\partial U_{l}}{\partial y}+w_{l}\frac{\partial U_{l}}{\partial z}-fV_{l}\\ \;=gh_{l}\frac{\partial \zeta }{\partial x}-\frac{gh_{l}}{\rho_{0}}\frac{\partial }{\partial x}\int_{-H_{l}}^{\zeta }\rho^{\prime }dz+\frac{h_{l}}{\rho_{0}}\frac{\partial p_{a}}{\partial x}+\frac{1}{\rho_{0}}\left(\tau_{x}^{top\left(l\right)}-\tau_{x}^{bottom\left(l\right)}\right)\\ \;+A_{H}\left(\frac{\partial^{2}U_{l}}{\partial x^{2}}+\frac{\partial^{2}U_{l}}{\partial y^{2}}\right)+\frac{\partial }{\partial z}\left(\frac{K_{l}}{h_{l}}\frac{\partial U_{l}}{\partial z}\right)\\ \frac{\partial V_{l}}{\partial t}+u_{l}\frac{\partial V_{l}}{\partial x}+v_{l}\frac{\partial V_{l}}{\partial y}+w_{l}\frac{\partial U_{l}}{\partial z}-fU_{l}\\ \;=gh_{l}\frac{\partial \zeta }{\partial y}-\frac{gh_{l}}{\rho_{0}}\frac{\partial }{\partial y}\int_{-H_{l}}^{\zeta }\rho^{\prime }dz+\frac{h_{l}}{\rho_{0}}\frac{\partial p_{a}}{\partial y}+\frac{1}{\rho_{0}}\left(\tau_{y}^{top\left(l\right)}-\tau_{y}^{bottom\left(l\right)}\right)\\ \;+A_{H}\left(\frac{\partial^{2}V_{l}}{\partial x^{2}}+\frac{\partial^{2}V_{l}}{\partial y^{2}}\right)+\frac{\partial }{\partial z}\left(\frac{K_{l}}{h_{l}}\frac{\partial V_{l}}{\partial z}\right)\\ \frac{\partial \zeta }{\partial t}+\sum_{l}\frac{\partial U_{l}}{\partial x}+\sum_{l}\frac{\partial V_{l}}{\partial y}=0 \end{array}$$
where l indicates the vertical layer, ($U_{l},V_{l}$) the horizontal transport components in *x* and *y* directions for each layer, $(u_{l},v_{l},\;w_{l})\;$ the velocity components, f the Coriolis parameter, $p_{a}\;$ the atmospheric pressure, g the gravitational constant, ζ  the sea level, $\rho_{0}\;$ the standard water density, $\rho =\rho^{\prime }+\rho_{0}\;$ the water density, τ  the internal stress term at the top and bottom of each layer, $h_{l}\;$ the layer thickness, $H_{l}$ the depth of the bottom of the layer l, and $A_{H}\;$ the horizontal eddy viscosity. The GOTM (General Ocean Turbulence Model) turbulence closure model described in Burchard et al. [58] was used for the computation of the vertical viscosity, $K_{l}$. Momentum exchanges across the model layers are accounted for by computing both the advective contribution and the vertical constituents of the diffusive terms $\frac{\partial }{\partial z}\left(\frac{K_{l}}{h_{l}}\frac{\partial U_{l}}{\partial z}\right)\;and\;\frac{\partial }{\partial z}\left(\frac{K_{l}}{h_{l}}\frac{\partial V_{l}}{\partial z}\right)$. Wind and bottom friction terms corresponding to the boundary conditions of the stress terms ($\tau_{x},\tau_{y}$) are defined as
(2)$$\begin{array}{l} \tau_{x}^{surface}=c_{D}\rho_{a}wi_{x}\sqrt{wi_{x}^{2}+wi_{y}^{2}}\\ \tau_{x}^{bottom}=c_{B}\rho_{0}u_{L}\sqrt{u_{L}^{2}+v_{L}^{2}}\\ \tau_{y}^{surface}=c_{D}\rho_{a}wi_{y}\sqrt{wi_{y}^{2}+wi_{y}^{2}}\\ \tau_{y}^{bottom}=c_{B}\rho_{0}v_{L}\sqrt{u_{L}^{2}+v_{L}^{2}} \end{array}$$
with $c_{D}$ as the wind drag coefficient, $c_{B}$ the bottom friction coefficient, $\rho_{a}$ the air density, ($wi_{x},\;wi_{y}$) the wind velocity components, and ($u_{L},v_{L}$) the bottom velocity components.

The hydrodynamic module was coupled with a solute transport model to compute the spreading and the fate of a Eulerian conservative tracer. Details of the model equations and adopted numerical solution are reported in Umgiesser et al. [51].

### 4.2. Model Setup

In previous studies by Denaro et al. [19], SHYFEM has been applied to reproduce the 3D current fields in the Augusta Bay and harbor as induced by the main meteorological and oceanographic forcing, including atmospheric pressure gradients and heat fluxes, winds, and tides, during a ten-year period between 2007 and 2017. In that study, the data obtained by the hydrodynamic model run were adopted as inputs by an off-line advection–diffusion and biogeochemical model based on a regular mesh that was implemented to simulate the fate of Hg in the Augusta Bay. The surface current fields obtained using the hydrodynamic model application were validated throughout the comparison with the results obtained by previous similar numerical applications carried out in the same area by Denaro et al. [19]. Furthermore, in Canu et al. [47], SHYFEM was applied to reproduce at high resolution the 3D water circulation and the fate of an oil spill at sea in the Sicilian coastal waters, including Augusta Bay. Finally, Cucco et al. [18] followed a similar approach to reproduce in the three dimensions and with high detail the water circulation and the dynamics of the water temperature and salinity in the Messina Strait and in the surrounding coastal waters, including the eastern Sicilian Shelf, where Augusta Bay is located.

In this work, SHYFEM was used to simulate the spread of Hg from Augusta Bay into the adjacent coastal areas under perturbation scenarios corresponding to (i) present conditions (Scenario 1) and (ii) lack of seawalls in the event of an earthquake (Scenario 2). No intermediate situation (partially breached seawalls) is discussed here.

The same model mesh used in Denaro et al. [19] was adopted for the Scenario 1 simulation run; this mesh is constituted by a finite element computational grid composed of 21,379 nodes and 40,486 triangular elements and extending between 15.05° E and 15.55° E and between 36.95° N and 37.35° N, with a spatial resolution varying between 20 m for the inner harbor and several km for the far field. Bathymetric details were reproduced using digitalized nautical charts for the inner bay and coastal areas along with the GEBCO dataset (https://www.gebco.net/data_and_products/gridded_bathymetry_data/, accessed on 15 July 2021) for the outer domain. In Figure 4, part of the finite element mesh and bathymetric details representing the model domain adopted in the Scenario 1 are reported (panel a and panel b).

For Scenario 2, the previously described model mesh was modified by adding new computational elements in the areas corresponding to the seawalls to reproduce a passage generated by the collapse of the dams. New bathymetric details were therefore added, reconstructing the morphological features of the collapsed seawalls by setting the water depth of each new element equal to the 50% of the averaged water depths obtained from the surrounding computational elements. This approach was justified by hypothesizing the total destruction of the seawalls, with consequent remobilizing and seaward dispersal of the pollutants. In Figure 5, the zoom of the bathymetric details for Augusta Harbour (panel a) and the three seawall areas (panels b for the southern seawall, c for the central seawall and d for the northern seawall) are reported as reproduced by the computational mesh adopted in the Scenario 2.

For both scenarios, the model vertical direction was defined by 22 z-levels, with layer depths ranging between 5 m and 200 m following an ad hoc step distribution.

Starting from previously-used modelling [18,47], we considered all the main forcing variables, including tides, wind, and thermohaline contributions, in reproducing the water circulation inside the harbor, in the adjacent coastal area, and in the offshore part of the domain. In particular, hourly fields of atmospheric forcing, including wind, precipitation, and thermal flux data provided by the meteorological prediction system SKIRON [59] were used as model surface boundary conditions. For both scenarios, oceanographic boundary conditions, including daily time series of water temperature, salinity, and water levels obtained from the reanalysis of the Mediterranean Forecasting System (MFS) [59] released by the Copernicus platform (http://marine.copernicus.eu, accessed on September 2020) and tidal data obtained for the whole considered period from the global tidal model OTIS (http://volkov.oce.orst.edu/tides/otis.html, accessed on 1 September 2020), were imposed along the model open boundary corresponding to the open sea mesh border. For all scenarios, simulations referred to the period between 1 January 2014 and 31 December 2014, corresponding to the same year simulated in the previous study [18,19]. The same model parameterization adopted in Cucco et al. [18] and Danaro et al. [19], applying SHYFEM with success to Augusta Bay and the surrounding coastal area and to the Eastern Sicilian shelf and coastal areas, was applied.

### 4.3. Scenario Setup

The numerical model was applied investigating the least probable and most impactful event, with the whole seawall system suddenly and entirely destroyed by an earthquake. In fact, we followed a simplified approach to reproduce the harbor water spillover in order to provide a semi-quantitative evaluation of the possible risk for pollution, as the event, though strongly unpredictable, would certainly be catastrophic. Nevertheless, the proposed method could be applied to infer and analyse the consequences of less catastrophic events with only a part or small sections of the seawall collapsed.

A full-year simulation run was carried out to reproduce the water circulation in the two scenarios. The risk of contamination was evaluated by reproducing the advection and diffusion of a passive tracer (Hg_T_ in the specific exercise) continuously released in the bay waters. This approach was based on the assumption of a dynamic equilibrium between Hg concentrations in the seafloor sediments and in the water column and of steady-state conditions for Hg in the seawater following the experimental/modelling results achieved for the study area. Specifically, the Hg_T_ concentration in the bay waters was set as nearly constant and its spatial distribution was defined from measurements, assuming the bottom sediments as a steady and unlimited source of Hg and a steady equilibrium between its content in the water and sea floor. Thus, the Hg_T_ concentration in seawater within the bay was set as nearly constant [42] and with median values of 17.9, 14.9, and 9.17 ng L^−1^ for the bottom, intermediate, and surface waters, respectively. The advection and diffusion of the tracer concentration was simulated considering the same steady distribution for both scenarios, with the setup differing only for bathymetric and geometric details in the seawall area.

## 5. Results

In the modelling run, the main assumption is that the Hg dispersion dynamic is treated as a sequence of equilibrium states which are quickly reached if compared with the time scale of the simulation run, which is one year. Within this context, the model results can be evaluated only in terms of yearly averaged distributions without any possible inference as to the intra-annual variability.

The simulation results, consisting of three-dimensional hourly fields of the tracer concentration obtained for the whole simulated period, allow evaluation of the potential impact of a massive release of Hg in the coastal waters after destruction of the seawall system confining Augusta Bay and for the year following the initial destruction. Figure 6 reports the results for the current situation and for the post-earthquake water mass circulation obtained for the surface and the bottom layers as well as for the averages computed along the water column.

In scenario 1, the yearly surface average Hg concentration is higher in the southern part of Augusta Bay close to the southern mouth, and lower in the northern part, where the outflow is quickly driven southward by the dominant coastal circulation. In Scenario 2, the same circulation now results in elevated concentrations extending outside of the bay, while concentrations within the bay reach higher levels. In this case, the outflow from the harbor boundaries increases the spreading of the tracer and its average concentration values both north and south of the bay. Similar results are found for the bottom, accompanied by increments of both the average concentration values of the tracer and the total impacted area as compared to scenario 1.

The differences between the average tracer concentrations are well described by the left panels of Figure 6, evidencing a high impact of the tracer outflow through the collapsed seawalls, mainly corresponding to the coastal areas and specifically in front of the northern and central dams, where the water is generally deeper. On the contrary, in front of the harbor inlets and in the off-shore areas the differences are mostly negligible. These differences can be observed for both the surface and bottom layers and for the vertical average distribution.

For both the proposed scenarios and for each element of the model domain, the maximum values of Hg concentration computed during the whole simulation run were detected and plotted (Figure 7) for both the surface and bottom layers. The obtained spatial distributions allow for detection of the maximum limit of the spreading area of Hg during the whole simulation run.

Contrary to the time-averaged distribution (see Figure 6), the analysis of the extremes reveals that during the simulation run as well as in both scenarios Hg spreads at the surface, even in the open sea. In fact, even if the maximum values are always found in proximity to the coast and south of the bay, a broad extent of the outer and northern part of the model domain shows high Hg surface concentration values. On the other hand, in both scenarios the maximum values in the bottom layers are found in proximity to the coast and mainly southward.

The differences in the maximum tracer concentration between the two scenarios are depicted in the left panels of Figure 6. As opposed to the time-averaged distribution, in this case the highest discrepancies are found off-shore for the surface distribution and near the coast for the bottom tracer concentration. A possible explanation is that in Scenario 2 the water masses exiting through the northern collapsed dam are intercepted by the offshore current, which promotes the transport of the tracer southward and offshore. In Scenario 1 this process is less pronounced, as the outflow through the northern inlet is less influenced by the offshore current. On the contrary, the outflow through the collapsed central dam dominates the tracer distribution at the seafloor, determining the greatest differences between the two scenarios in the coastal areas in front of this dams. Finally, for both surface and bottom layers little remarkable difference was found in the coastal area in front of the southern dam, as the outflow of the tracer in this area was limited to the few meters at the surface in the case of Scenario 2, and therefore dominated by the inflow and outflow through the southern inlet.

In Table 1 these differences are quantified in terms of the total surfaces of the model domain which experienced Hg concentration values higher than 1 during the simulation run. Unitary tracer concentration values were arbitrarily selected as references for comparing the results of the two scenarios. For both scenarios, the values are expressed in m^2^ and are averaged over time.

This approach was followed to perform a relative evaluation of the impact of seawall collapse on the Hg distribution outside and inside the bay. In fact, even if arbitrarily selected, the definition of a common threshold allows estimation of the relative increment of the contamination risk through a direct comparison between the results of the two scenarios. For an absolute evaluation of the impacts, in addition to the advection and diffusion processes all of the biogeochemical reactions affecting Hg availability in the water column and sediment should be considered in the numerical simulation [19]. This was not the focus of the present work, which aimed to provide an initial helpful evaluation of the relative increment of Hg dispersion in the investigated area following a catastrophic event.

In Scenario 2, both the yearly average surfaces are always higher than in Scenario 1, with a calculated average outflow of 0.075 kmol/y^−1^ [19] for both the surface and bottom layers and for both average and maximum distributions. In particular, for the average distribution the relative differences between the two scenarios results reveal that the area of the model domain with Hg concentration values higher than 1 is 47% larger for the surface layer and about 25% larger for the bottom layer in Scenario 2 compared to Scenario 1. Similarly, for the maximum tracer concentration values, the total surface area with tracer concentration higher than 1 is more extensive in scenario 2 both at the surface (82% larger, corresponding to a total of 0.11 kmol/y^−1^) and on the bottom (40% larger, corresponding to a total of 0.093 kmol/y^−1^).

## 6. Discussion

In this section, the results of the new model are analysed considering the maximum values of the Hg concentration, with the objective of detecting and investigating the areas potentially subjected to extreme effects. To explore the effect of this extra Hg outflow at the Mediterranean scale we used the steady-state mass balance box model proposed by Salvagio Manta et al. [42] for Augusta Bay and re-calculated by Denaro et al. [19]. The results of Scenario 2 imply an increase in the rate of Hg input to the Mediterranean from its current value of less than 3–4% to a new value of 7–8% of the total. This increase would occur almost instantaneously relative to the simulation run time of the model. It is worth noting that the above estimate is a conservative one, as it does not consider specific extra inputs of Hg related to potential sediment resuspension due to earthquake activity in an area with high Hg concentrations.

The simulated effects of dispersion of contaminants in seawater produced by the impact of an earthquake on the seawall confining Augusta Bay are particularly significant in terms of the distribution pattern compared to the present conditions. These simulations prompt the need to include any potential consequence induced or amplified by geological hazards in the risk assessment of any specific area already impacted by environmental contamination. This is a change in systemic approach from a static and a single-hazard approach (specifically due to the potential effects of contaminants on the ecosystem and human health via food web transfer and fish diet) to a dynamic and multi-hazard view. Indeed, a multi-hazard effect induced by amplification/combination of risks associated with the dynamics of specific contaminated environmental compartments randomly connected to geo-hazards should be part of a contemporary and more appropriate approach to environmental risk assessment. In particular, sensitive areas where seismogenic and/or tsunamigenic geological components potentially increase hazard factors to environmental systems where the biogeochemical dynamics of contaminants, per se, already pose a risk to the coastal and marine ecosystem. The disaster at Fukushima Daiichi and the destructive swell at Rapallo represent two paradigmatic case studies underlining the necessity of approaching modern risk assessment through multi-hazard analysis. This appears much more manifest in sensitive areas where the risk to the ecosystems posed by the biogeochemical cycling of contaminants is further amplified by tectonic hazards. The environmental/geological setting considered here is a case in point, where (i) a number of highly contaminated landfills occur as specific legacy of industrial activity, (ii) inappropriate management has produced contamination in several environmental matrices, and (iii) a geological multi-hazard represents a specific threat to terrestrial, coastal, and marine systems. Impacts from enhanced and larger-scale spreading of contaminants in relation to their biogeochemical dynamics have to be estimated in order to effectively evaluate the risk associated with highly contaminated areas.

## 7. Conclusions

We explored the impact of an earthquake on human artifacts (in this specific case, the sudden breaching and complete collapse of the system damming the heavily polluted Augusta Bay) and the resulting dispersion of mercury (treated as an example of the various contaminants documented in the environmental matrices of this study area) at a large scale in an exercise of multi-hazard impact on a highly polluted marine coastal area. The simulated significant increase of the Hg point-source input to the Mediterranean from its current levels to peaks of 7–8% of the total, corresponding to an average of 0.241 kmol y^−1^, occurring almost instantaneously relative to the simulated collapse, suggest that natural and artificial hazards are inextricably connected and demand innovative methodological approaches for planning appropriate risk reduction and policy management.

The achieved results clearly suggest that highly polluted landfills and/or coastal areas where major anthropogenic contamination has resulted from intensive industrial activities solicit a major effort and specific consideration in terms of integrated methodological approaches for appropriate evaluation of multi-hazards associated with natural events impacting on the biogeochemical dynamics and spreading pollutants in the marine environment on highly variable temporal and spatial scales. Within this context, future implementation and application of a fully coupled oceanographic and biogeochemical numerical model is mandatory in seeking to deepen and quantify the hazard and risk of pollution events related to such extreme events.

## Figures and Tables

**Figure 2 ijerph-19-03956-f002:**
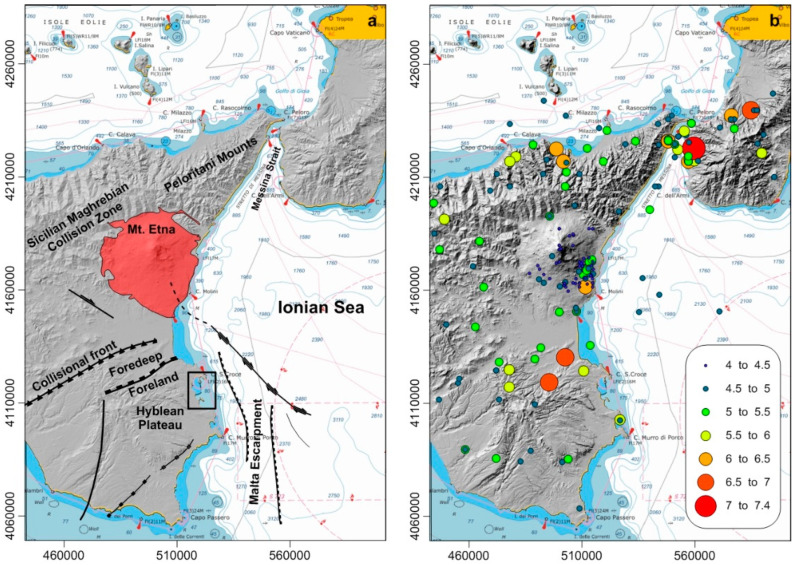
The area of Eastern Sicily with the background of main geological features (**a**), redrawn after [23] and historical instrumental epicenters (Seismic Magnitude) of Eastern Sicily, extracted by the CPTI04 INGV seismic catalogue (**b**).

**Figure 3 ijerph-19-03956-f003:**
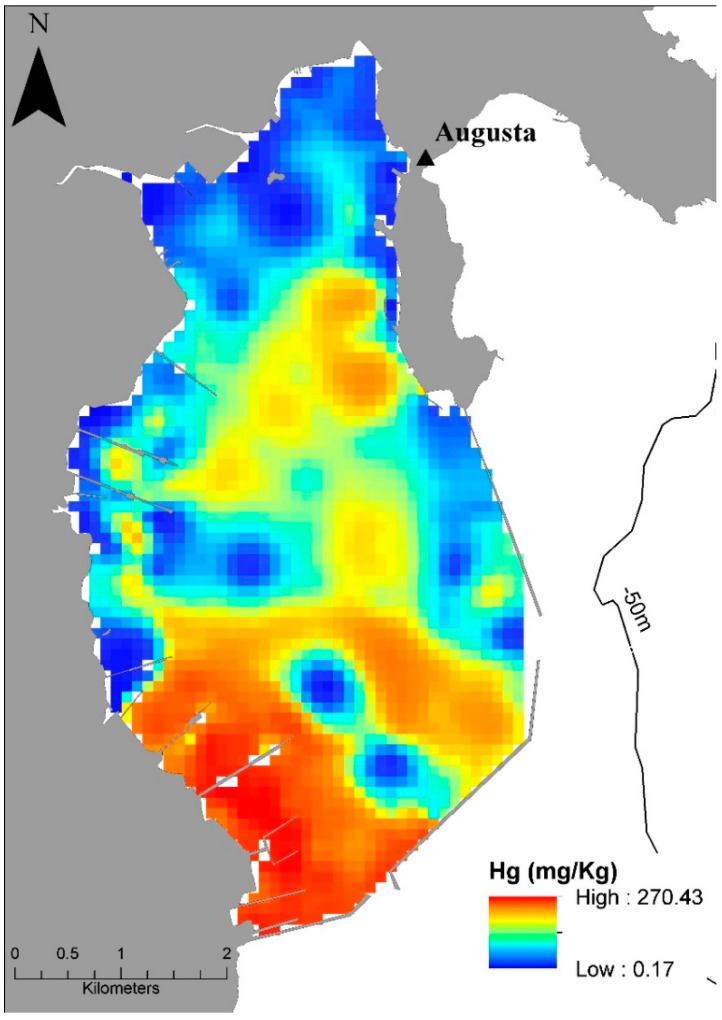
Distribution map of Hg in the surface (0–10 cm) sediments of Augusta Bay.

**Figure 4 ijerph-19-03956-f004:**
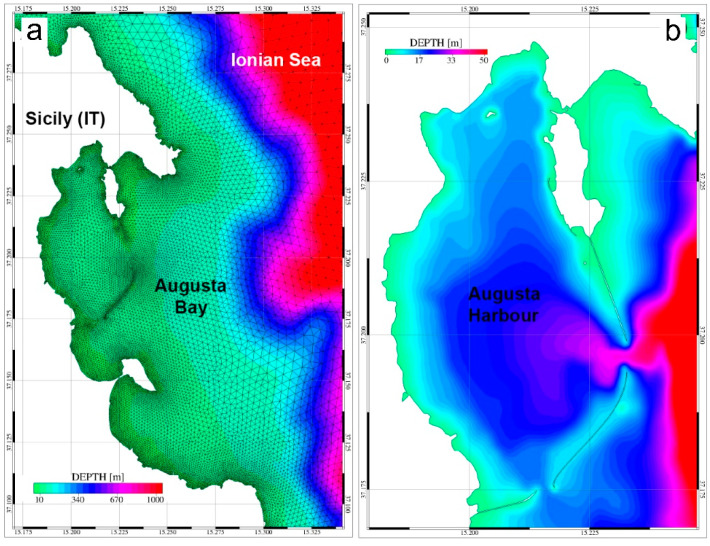
Model domain and bathymetric details. Part of the finite element mesh adopted to reproduce the geometry of Augusta Bay and the surrounding coastal areas (panel (**a**)). Domain geometry and bathymetric details for Augusta Harbour as adopted for the Scenario 1 (panel (**b**)).

**Figure 5 ijerph-19-03956-f005:**
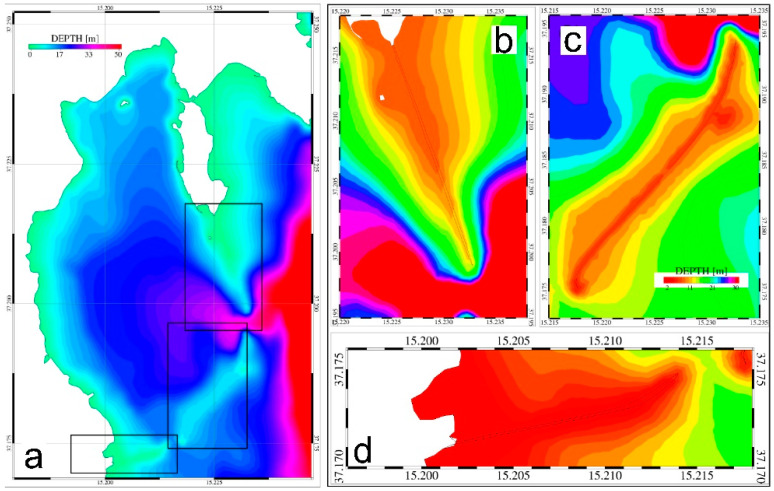
Domain geometry and bathymetric details for Augusta Harbour (**a**) and the collapsed seawall areas. Panel (**b**) depicts the northern seawall, panel (**c**) the central seawall, and panel (**d**) the southern seawall after the earthquake event adopted for Scenario 2.

**Figure 6 ijerph-19-03956-f006:**
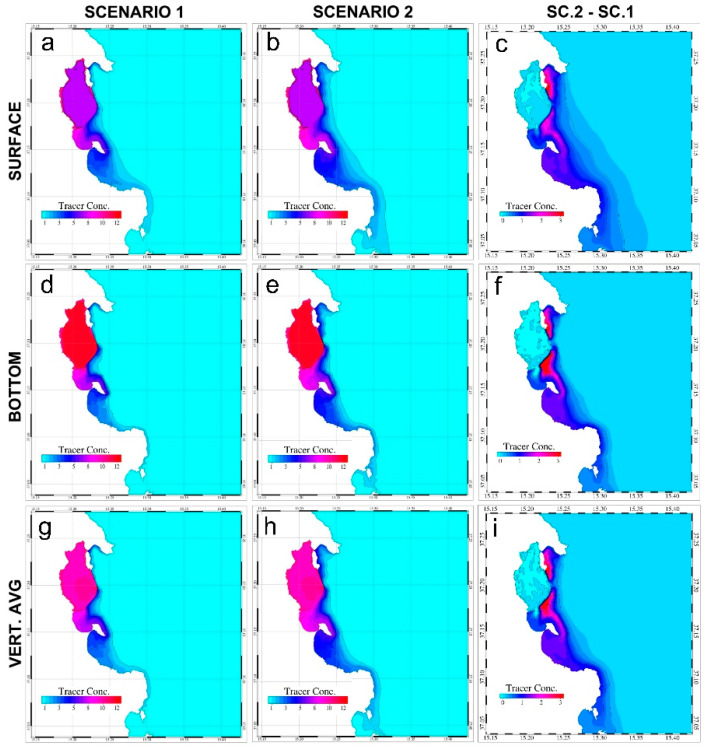
Yearly average distributions of tracer concentration values (mg L^−1^), computed for both scenarios for the surface layer (surface, panels (**a**,**b**)), the bottom layer (bottom, panels (**d**,**e**)), and the averages along the water column (vert. avg, panels (**g**,**h**)). The differences between the tracer concentration in Scenario 2 and Scenario 1 are reported in the left panels for the surface (panel (**c**)) and bottom layers (panel (**f**)) and for the vertical averages (panel (**i**)).

**Figure 7 ijerph-19-03956-f007:**
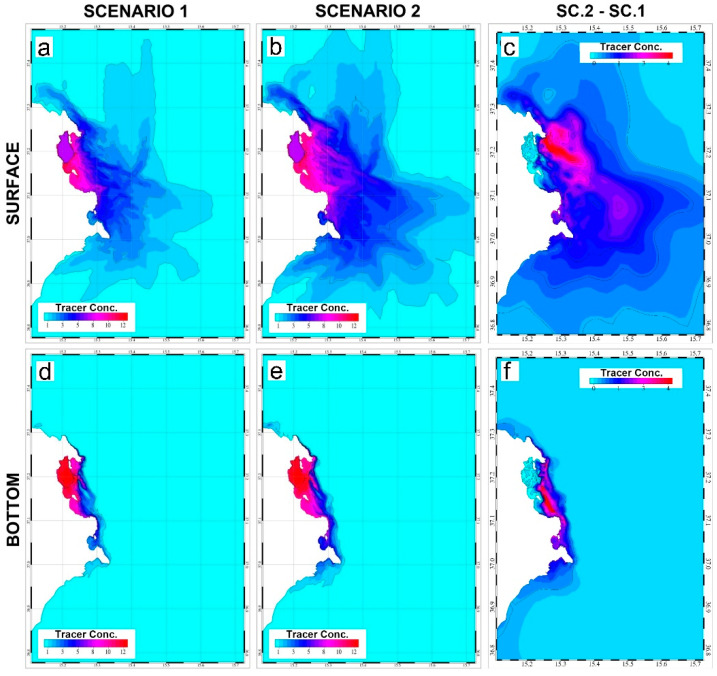
Spatial distribution of the maximum tracer concentration values (mg L^−1^) computed during the whole simulation run for each element of the model domain for both scenarios at the surface (panels (**a**,**b**)) and bottom layers (panels (**d**,**e**)). The differences between the maximum tracer concentration in Scenario 2 and Scenario 1 are reported in the left panels for surface (panel (**c**)) and bottom layers (panel (**f**)).

**Table 1 ijerph-19-03956-t001:** Differences between the extent of the tracer distributions in the two modelled scenarios. Top: surfaces of the model domain (expressed in m^2^) characterized by yearly mean values of tracer concentration. Bottom: surfaces of the model domain (expressed in m^2^) that experienced maximum tracer concentration values during the whole simulation run. “*Q*” indicates the relative differences between the areas obtained for the two scenarios.

	SURFACES WITH AVG TRACER CONC. > 1		
	*SCENARIO 1*	*SCENARIO 2*	*Q*
SURFACE	6.96 × 10^7^	1.02 × 10^8^	47%
BOTTOM	4.54 × 10^7^	5.69 × 10^7^	25%
	SURFACES WITH MAX TRACER CONC. > 1		
SURFACE	1.33 × 10^9^	2.42 × 10^9^	82%
BOTTOM	1.38 × 10^8^	1.94 × 10^8^	40%

## Data Availability

Not applicable.

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
