# Peer review of "Large-Scale Mercury Dispersion at Sea: Modelling a Multi-Hazard Case Study from Augusta Bay (Central Mediterranean Sea)"

_ijerph, 2022, doi:10.3390/ijerph19073956_

Round 1

Reviewer 1 Report

In this paper, the authors presented the analysis of Large-Scale Mercury Dispersion at Sea: Modelling a Multi-Hazard Case Study from Augusta Bay (Central Mediterranean Sea). This paper has a certain degree of novelty and has carried out specific research on the research object, however there are still some questions in this manuscript. The following are the questions in this manuscript:

  • The text is not well arranged and the logic is not clear.
  • In page 7 line 268, the authors mentioned that “as induced by the main meteorological and oceanographic forcing during a ten years period between 2007 and 2017 ”,there is a lack of explanation of “the main meteorological and oceanographic forcing”.
  • In page 8 line 300, the authors mentioned that “This approach was justified by hypothesizing the total destruction of the seawalls with a partial removal of the debris by the Tsunami wave impact ”,there is a lack of specific explanation of rationality.
  • In page 12 line 420, the authors mentioned that “Unitary tracer concentration values were arbitrarily selected as references for the comparison of the two scenarios results”,too little data may not be convincing.
  • 4 in page 9 shows model domain and bathymetric details, but the pictures shown in the paper are too small and not sufficiently clear.

To sum up, I suggest that this manuscript need minor revision.

Author Response

Reviewer#1

In this paper, the authors presented the analysis of Large-Scale Mercury Dispersion at Sea: Modelling a Multi-Hazard Case Study from Augusta Bay (Central Mediterranean Sea). This paper has a certain degree of novelty and has carried out specific research on the research object, however there are still some questions in this manuscript. The following are the questions in this manuscript:

  • The text is not well arranged and the logic is not clear.
  • In page 7 line 268, the authors mentioned that “as induced by the main meteorological and oceanographic forcing during a ten years period between 2007 and 2017 ”,there is a lack of explanation of “the main meteorological and oceanographic forcing”.

Answer: We changed the original text “In previous studies [17], SHYFEM has been applied to reproduce the 3D current fields in the Augusta bay and harbor, as induced by the main meteorological and oceanographic forcing during a ten years period between 2007 and 2017.”

with the following:

In previous studies [17], SHYFEM has been applied to reproduce the 3D current fields in the Augusta bay and harbor, as induced by the main meteorological and oceanographic forcing, including atmospheric pressure gradients and heat fluxes, winds and tides, during a ten years period between 2007and 2017.

  • In page 8 line 300, the authors mentioned that “This approach was justified by hypothesizing the total destruction of the seawalls with a partial removal of the debris by the Tsunami wave impact ”,there is a lack of specific explanation of rationality.

Answer: We changed the original text “This approach was justified by hypothesizing the total destruction of the seawalls with a partial removal of the debris by the Tsunami wave impact.”

With the following:

“This approach was justified by hypothesizing the total destruction of the seawalls with consequent remobilizing and seaward dispersal of the pollutants.”

  • In page 12 line 420, the authors mentioned that “Unitary tracer concentration values were arbitrarily selected as references for the comparison of the two scenarios results”,too little data may not be convincing.

Unitary tracer concentration values were arbitrarily selected as references for the comparison of the two scenarios results. For both scenarios the values are expressed in m2 and are averaged over time.

Answer: We changed the original text “Unitary tracer concentration values were arbitrarily selected as references for the comparison of the two scenarios results. For both scenarios the values are expressed in m2 and are averaged over time.

With the following:

Unitary tracer concentration values were arbitrarily selected as references for the comparison of the two scenarios results. For both scenarios the values are expressed in m2 and are averaged over time.

This approach was followed to perform a relative evaluation of the impact of the seawalls collapse on the Hg distribution outside and inside the bay. In fact, even if arbitrarily selected, the definition of a common threshold allows to estimate the relative increment of the contamination risk trough out a direct comparison between the two scenarios results. For an absolute evaluation of the impacts, in addition to the advection and diffusion processes, all the biogeochemical reactions affecting the Hg availability in the water column and sediment should be considered in the numerical simulation [17]. This was not the focus of the present work which aimed at providing a first but helpful evaluation of the relative increment of the Hg dispersion in the investigated area after a catastrophic event.4 in page 9 shows model domain and bathymetric details, but the pictures shown in the paper are too small and not sufficiently clear.

  • 4 in page 9 shows model domain and bathymetric details, but the pictures shown in the paper are too small and not sufficiently clear.

Answer: Figure 4 has been splitted into two new figures, the 4th and the 5th, in order to improve the quality of the images. In the revised version, the new figure 4 describes part of the finite element mesh and the bathymetric details of the Augusta bay and harbour as used for Scenario 1. In the new figure 5, the geometry and bathymetry of the Augusta harbour and of the collapsed sea walls, as reproduced by the model mesh for the Scenario 2, are depicted.

To sum up, I suggest that this manuscript need minor revision.

Answer: We thank the reviewer#1 for the precise and very fast review of our paper.

Reviewer 2 Report

Highlight changes in yellow in a next revision, please. No track changes.

The similarity present in the text should be reduced, not considering content used from another native language…

, particularly in specific cases where it is extensive

There is known content without the respective citation.

That is serious.

There is no need to use sentences already published

Each time known equations (distorted in the PDF and to be kept to the minimum)/data/information are mentioned, the references must be immediately present before.

All parameters, and units wherever available in “()” must be defined after

General comments:

Change language: “

 Abstract: This contribution discusses”

Start by brief contextualization, then methods, main findings and practical implications

This single image could be moved to the main text:

“Fig. S1 in supplementary materials”

Do not use upper letter to identify separate images

Why include methods/objectives, in the middle?

“this work we simulate the effects of a potential whole collapse of the Augusta sea-86 walls (the artificial breakwater system enclosing the bay) under external stresses. 87

Such a kind of extreme event could occur in the eastern Sicily coasts that underwent 88 a number of seismic events considered among the strongest ever observed in Italy in his-89 torical past (e.g., [10]), and also occasionally generated a tsunami ([11] and”

Restructure content insertion

Assure figures are original, otherwise remove.

See that a  min caption needs to be present:

Figure 1. A) Location Map of the Augusta Bay. B) Coastal toponyms of the Augusta Bay and of the 125 Rada of Augusta. Bathymetry in B modified after.”

Plural? Then more: “In previous studies [17],”refer to the authors names directly in such cases…

And check all “The same model mesh used in [17],”

“re-calculated by [17].”

Etc

etc

AS in other cases, grouped figures need to be identified and subcaptions added…

Figure 6. spatial distribution of the maximum tracer concentration values (mgl-1) computed during 400 the whole simulation run for each element of the model domain and for both scenarios. The differ-401 ences between the maximum tracer concentration in Scenario 2 and Scenario 1 are reported in the 402 left panels for surface and bottom layers.”

Why aggressive upper letter?

Table 1. Differences between the extent of the tracer distributions in the 2 modelled Scenarios. Top: 433 surfaces of the model domain expressed in m2 characterized by yearly mean values of the tracer 434 concentration. Bottom: surfaces of the model domain, expressed in m2, that experienced maximum 435 tracer concentration values during the whole simulation run. Q indicates the relative differences 436 between the areas obtained for the two scenarios.”

Check the italics to parameters: “Q”: text…

Conclusions: use the above suggested structure to the abstract /same structure, different content):

Contextualization, etc

Clarify methods

Highlight innovation and novelty

Add limitations

The conclusions are just too general

See that the references should include more international regional authors

Add more references from 2022

I hope the authors are able to further contribute to a more relevant text

Author Response

Reviewer#2

The similarity present in the text should be reduced, not considering content used from another native language…, particularly in specific cases where it is extensive

Answer: We checked the text and tried to verify and improve the document and making it original in all its parts.

There is known content without the respective citation. That is serious.

Answer: We checked the text and verified that all information from other authors are  properly cited.

There is no need to use sentences already published

Answer: All sentences are original. Some of them not reporting scientific novelties (e.g., about the description of the state of the art), now mention specific references.

Each time known equations (distorted in the PDF and to be kept to the minimum)/data/information are mentioned, the references must be immediately present before.

Answer:  We checked the manuscript and now all the equations/data/information are correctly mentioned before or after they are taken into account for the flow of the discussion.

All parameters, and units wherever available in “()” must be defined after

Answer: Done

General comments:

Change language: “Abstract: This contribution discusses”. Start by brief contextualization, then methods, main findings and practical implications

Answer: We rephrased a little bit the Abstract and arranged it following the suggestion.

This single image could be moved to the main text:“Fig. S1 in supplementary materials”

Answer: Frankly, we don’t think the proposed solution could significantly improve the readability of the manuscript and prefer to maintain the S1 in the supplementary material.  this really mandatory

Do not use upper letter to identify separate images

Answer: Done

Why include methods/objectives, in the middle?

“this work we simulate the effects of a potential whole collapse of the Augusta sea-86 walls (the artificial breakwater system enclosing the bay) under external stresses. 87 Such a kind of extreme event could occur in the eastern Sicily coasts that underwent 88 a number of seismic events considered among the strongest ever observed in Italy in his-89 torical past (e.g., [10]), and also occasionally generated a tsunami ([11] and”Restructure content insertion

Answer: We rephrased to make it much the content more understandable

 Assure figures are original, otherwise remove.

Answer: All figures are original

See that a  min caption needs to be present:

Answer: All figures are illustrated by a caption

Figure 1. A) Location Map of the Augusta Bay. B) Coastal toponyms of the Augusta Bay and of the 125 Rada of Augusta. Bathymetry in B modified after  

Plural?

Answer: Is not clear to us if the reviewer#2 refers to the term “toponyms”. In such a case, there are several toponyms in the frame 1B. However, we changed the original caption with the following:

Figure 1. A) Location Map of the Augusta Bay. B) Coastal frame of Augusta Bay and Rada of Augusta. Bathymetry in B modified after Firetto Carlino et al. [15].

Then more: “In previous studies [17],”refer to the authors names directly in such cases…

Answer: We followed the Environmental Research and Public Health template (please, see https://www.mdpi.com/journal/ijerph/instructions)

And check all “The same model mesh used in [17],”“re-calculated by [17].”

Answer: We followed the Environmental Research and Public Health template (please, see https://www.mdpi.com/journal/ijerph/instructions)

Answer: We modified all figures adding sub-captions and letters to identify each single panel

Figure 6. spatial distribution of the maximum tracer concentration values (mgl-1) computed during 400 the whole simulation run for each element of the model domain and for both scenarios. The differ-401 ences between the maximum tracer concentration in Scenario 2 and Scenario 1 are reported in the 402 left panels for surface and bottom layers.”

 “Table 1. Differences between the extent of the tracer distributions in the 2 modelled Scenarios. Top: 433 surfaces of the model domain expressed in m2 characterized by yearly mean values of the tracer 434 concentration. Bottom: surfaces of the model domain, expressed in m2, that experienced maximum 435 tracer concentration values during the whole simulation run. Q indicates the relative differences 436 between the areas obtained for the two scenarios.” Check the italics to parameters: “Q”: text…

Answer: Done

Conclusions: use the above suggested structure to the abstract /same structure, different content):

Contextualization, etc

Clarify methods

Highlight innovation and novelty

Add limitations

The conclusions are just too general

Answer: We improved the Conclusion paragraph following suggestions from the reviewer.

See that the references should include more international regional authors. Add more references from 2022.

Answer: Some more recent references (2020 and 2021) were added.

We thank the reviewer#2 for the very fast review of our paper.

Round 2

Reviewer 2 Report

Highlight changes in yellow in a next revision, please. No track changes.

Before the subcaptions…

“See that a 

amin caption needs to be present:

Answer: All figures are illustrated by a caption

I do not need to see the template. What I say is that is these cases, the authors names should be directly addressed…

Then more: “In previous studies [17],”refer to the authors names directly in such cases…

Answer: We followed the Environmental Research and Public Health template (please, see https://www.mdpi.com/journal/ijerph/instructions)”

“And check all “The same model mesh used in [17],”“re-calculated by [17].”

Answer: We followed the Environmental Research and Public Health template (please, see https://www.mdpi.com/journal/ijerph/instructions)

I clearly explained to the authors why the similarity issues are so important. I had to spare a significant amount of time to discover what I mentioned. Other than that, the paper has been improved. However, similarity is still present, and is significant in specific parts of the text, without references too

Please see that retraction watch cites more and more cases on similarity issues: https://retractionwatch.com/

Author Response

Reviewer#2

Comment: Highlight changes in yellow in a next revision, please. No track changes.

Answer: Ok

Before the subcaptions…

“See that a

amin caption needs to be present:

Answer: We checked that all figures are illustrated by a caption. Appropriate sub-captions were also added.

Comment:I do not need to see the template. What I say is that is these cases, the authors names should be directly addressed…

 “Then more: “In previous studies [17],”refer to the authors names directly in such cases…

Answer: done

“And check all “The same model mesh used in [17],”“re-calculated by [17].”

Answer: done

I clearly explained to the authors why the similarity issues are so important. I had to spare a significant amount of time to discover what I mentioned. Other than that, the paper has been improved. However, similarity is still present, and is significant in specific parts of the text, without references too

Please see that retraction watch cites more and more cases on similarity issues: https://retractionwatch.com/

Answer: We checked the text with “Grammarly for chrome” scan for plagiarism. We have detected no similarities. We remark that the text, the scientific idea, scientific data and the whole paper are original

Some (weak?) similarities may regard data and methods or the state of the art, but they are linked to citations.